

# IoT in urban development: insight into smart city applications, case studies, challenges, and future prospects

Sayeed Salih[1], Abdelzahir Abdelmaboud[2], Omayma Husain[3,4], Abdelwahed Motwakel[1], Hashim Elshafie[5], Mahir Sharif[6,7] and Mosab Hamdan[8]

[1] Department of Management Information Systems, College of Business Administration in Hawtat Bani Tamim, Prince Sattam bin Abdulaziz University, Al-Kharj, Saudi Arabia
[2] Humanities Research Center, Sultan Qaboos University, Muscat, Oman
[3] Department of Information Technology, Ajman University, Ajman, United Arab Emirates
[4] Department of Computer Science, University of Khartoum, Khartoum, Sudan
[5] College of Computer Science, King Khalid University, Abha, Saudi Arabia
[6] Computer Science Department, Faculty of Computer Science Information Technology, Omdurman Islamic University, Omdurman, Sudan
[7] Computer and Self Development, Prince Sattam bin Abdelaziz University, Al-Kharj, Saudi Arabia
[8] School of Computing, National College of Ireland, Dublin, Ireland

Corresponding author
Abdelzahir Abdelmaboud,
aelnour@kku.edu.sa

## ABSTRACT

With the integration of Internet of Things (IoT) technology, smart cities possess the capability to advance their public transportation modalities, address prevalent traffic congestion challenges, refine infrastructure, and optimize communication frameworks, thereby augmenting their progression towards heightened urbanization. Through the integration of sensors, cell phones, artificial intelligence (AI), data analytics, and cloud computing, smart cities worldwide are evolving to be more efficient, productive, and responsive to their residents' needs. While the promise of smart cities has been marked over the past decade, notable challenges, especially in the realm of security, threaten their optimal realization. This research provides a comprehensive survey on IoT in smart cities. It focuses on the IoT-based smart city components. Moreover, it provides explanation for integrating different technologies with IoT for smart cities such as AI, sensing technologies, and networking technologies. Additionally, this study provides several case studies for smart cities. In addition, this study investigates the challenges of adopting IoT in smart cities and provides prevention methods for each challenge. Moreover, this study provides future directions for the upcoming researchers. It serves as a foundational guide for stakeholders and emphasizes the pressing need for a balanced integration of innovation and safety in the smart city landscape.

# INTRODUCTION

The Internet of Things (IoT) is a cutting-edge concept that links objects connected to the Internet to send, receive, and process data. It organizes the object-internet relationship to

enable it to perform vital functions and be controlled over the network (*Al Homssi et al., 2020*). The IoT is a vast network of gadgets linked to the Internet, such as smartphones, tablets, and virtually anything with a sensor embedded in the device (*e.g.*, automobiles, machinery in manufacturing plants, jet engines, oil drilling equipment, and electronic watches). These gadgets collect and exchange data *via* a network (*Kasat et al., 2023*). Infrastructure for information and communication technology (ICT) underpins the function of smart cities.

Many cities worldwide, such as Dubai, New York, Tokyo, Shanghai, and Amsterdam, have launched innovative city programs. In the next decade, smart city models will be extensively embraced and act as the cornerstone of municipal development plans. According to *Ali & Ali (2021)*, the operation of IoT is supported by three key elements. The first and the most essential element is the radio waves that connect devices to the Internet to convey information *via* technologies (*i.e.*, WiFi, Bluetooth, NFC, and radio frequency identifier (RFID)). The devices (*e.g.*, automatic door locks, motion/remote sensing sensors, and room lights) are the second element in IoT operation. The third element refers to the use of cloud services that gather and analyze data to make decisions or arrive at a conclusion. Cloud computing enables all parties to exchange and share data by allowing access from any personal computer, palm device, smartphone, or web portal.

Smart cities have many goals and priorities, which all revolve around three elements: an infrastructure for ICT, an intricately designed integrated management framework, and intelligent citizens (*Ali & Ali, 2021*). ICT infrastructure is a requirement for smart cities to succeed and for their services to be effective. Smart cities need to adhere to rigorous standards. These standards enable various systems to operate, integrate, and harmonize effectively. To allow the usage of smart gadgets, smart cities have a crucial role in educating their citizens on how to use them safely.

While numerous studies have investigated IoT applications in smart cities, most focus on isolated aspects such as AI integration, sensing technologies, or security concerns, rather than offering a holistic perspective on IoT-based smart city ecosystems (*Alavi et al., 2022*; *Esfandi et al., 2024*). Despite advancements in IoT-enabled urban solutions, security and governance challenges remain underexplored, particularly concerning policy frameworks and interoperability between various smart city infrastructures (*Ahmad et al., 2023*). Furthermore, many existing surveys lack a systematic analysis of case studies across different regions, overlooking key lessons that could inform policymakers and urban developers. While previous studies, on the other hand, have broadly explored IoT applications in smart cities. For example, *Ilyas (2021)*, *Priya Dharshini et al. (2022)* discuss the lack of a detailed examination of security vulnerabilities and mitigation. This study addresses these gaps by analyzing case studies of IoT implementation, focusing on identifying practical solutions to security and scalability challenges. Addressing these gaps requires a more comprehensive approach synthesising technological, regulatory, and practical insights to create resilient and sustainable innovative city ecosystems. Figure 1 below shows the growth of smart city adoption from 2015 to 2030.

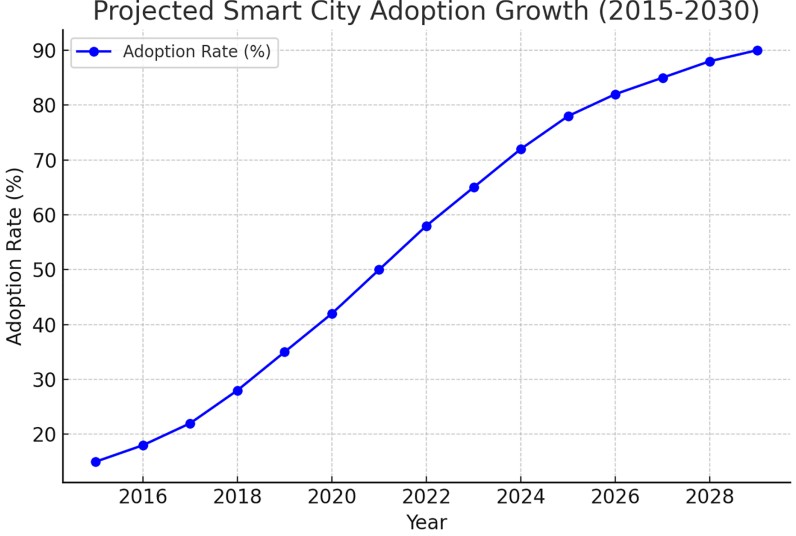

**Figure 1** **The primary components of IoT based smart city.**

## Motivation

The World Bank estimated that 54% of the world's population lived in cities, which was predicted to rise to 80% by 2045 (*UN, 2018*). The number of cities with a population of at least 10 million people rose from 10 in 1990 to 28 in 2014 and will reach 41 in 2030 (*United Nations Population Department of Economic and Social Affairs, 2014*). As many experts reported that future cities would collapse if did not become competent, it was anticipated that this number would increase between 80% and 100% in the Gulf Cooperation Council (GCC) countries (*Sanfilippo, Vermeersch & Benito, 2024*). The only solution in this case is to make the cities more effective, efficient, and rational by integrating innovative technology. Additionally, market city enterprises, according to *Kurniawan et al. (2023)*, would be worth USD 1 trillion by the end of 2030, thus providing a significant financial incentive for both government and private sector organizations to actively contribute to the progress of technology that facilitates the expansion of smart cities. With more than 80% of its population residing in cities, North America is the world's most populated region but has fallen behind in implementing smart cities (*Bauer, Sanchez & Song, 2021*). Thus, this is the main reason for studying the components of IoT-based smart cities and demonstrating the challenges of smart cities through the current use cases. Although many smart city solutions for public safety and transportation are already operational, some security concerns persist in IoT/IoT communication (*Bhardwaj et al., 2023*). These setbacks pose a significant challenge to the conventional urban infrastructure.

## Related studies

Many studies have explored IoT privacy and security risks. The work of *Chambers & Evans (2020)*, provided an overview of the system infrastructure for an urban IoT and emphasized the services commonly associated with the concept of "Smart City". The study

covered the associated data formats, communication protocols, connection layer technologies, and strategies of web service implementation. The study of *Javed et al. (2022)* highlighted the developing IoT concept and its vital role in establishing smart cities. The study stressed the importance of a cloud-oriented design that could integrate networks, software, sensors, user interfaces, and data analytics for value creation. In the work of *Rafique et al. (2020)*, edge computing was compared with typical cloud platforms to secure IoT networks. Similarly, *Farman et al. (2022)* studied the IoT-based smart transportation systems that use sensors, GPS, and AI to enhance traffic flow, reduce congestion, and improve public transportation efficiency. The results show that intelligent traffic management systems (ITMS) can minimize road congestion and fuel consumption by dynamically adjusting traffic lights and providing real-time routing information to drivers. The study contrasted the key elements and aspects of a smart city with IoT technology. The key contributions of past extensive studies on IoT-based smart cities are summarized in Table 1.

## Importance of the study

With rapid urbanization, cities worldwide face increasing infrastructure strain, traffic congestion, energy inefficiencies, and environmental concerns, necessitating innovative solutions for sustainable urban management. IoT technologies offer a transformative approach to optimizing city operations; however, existing research often remains fragmented, focusing on isolated applications such as smart transportation or energy management rather than providing a holistic analysis (*Oladimeji et al., 2023*). This study addresses this gap by comprehensively examining IoT-based smart city components, including AI integration, sensing technologies, wireless networks, and security considerations. While IoT-driven urban solutions enhance efficiency and sustainability, also introduce significant cybersecurity risks that many studies tend to overlook. This research highlights various IoT security vulnerabilities and proposes mitigation strategies to enhance the resilience of smart city infrastructures. By analyzing real-world implementations in cities such as Masdar City, New York, and Singapore, this study provides practical insights into the effectiveness of IoT applications in diverse urban contexts. Furthermore, it underscores the need for cross-country comparative studies to assess how economic and cultural factors influence the success of IoT-based smart city initiatives. Additionally, it emphasizes the importance of investigating IoT's role in promoting environmental sustainability, improving energy efficiency, and reducing carbon footprints. Given the increasing reliance on IoT for urban development, this study provides critical recommendations for policymakers, city planners, and researchers to enhance the security, privacy, and regulatory frameworks surrounding IoT adoption in smart cities.

## Contributions

This study is focused on the IoT-based innovative city components. Moreover, it explains how to integrate different technologies with IoT for smart cities, such as AI, sensing technologies, and networking technologies. Additionally, this study offers several case

**Table 1  Past research articles on the IoT of smart cities.**

| No. | Year | Author | Contribution and results |
|---|---|---|---|
| 1 | 2020 | Manickam & Kooy (2020) | The study investigated security and privacy challenges related to IoT adoption for smart cities. |
| 2 | 2020 | Andrade et al. (2020) | A top-down approach was used to assess cybersecurity incident response for IoT ecosystems. A paradigm predicated on risk stratification was built and tested to assess IoT cybersecurity maturity in smart city. |
| 3 | 2020 | Cvar et al. (2020) | The article presents a summary of technological options that assist intelligent solutions in smart cities and smart villages. |
| 3 | 2020 | Ahad et al. (2020) | The study discusses the obstacles to smart city development and the mitigating solutions. Three issues were detected: technological, socioeconomic, and environmental problems. |
| 4 | 2021 | Syed et al. (2021) | A review of the most popular applications in various fields of smart city research work. It analyzed the challenges of IoT systems for smart city encounters and the mitigating solutions. |
| 5 | 2021 | Bauer, Sanchez & Song (2021) | This study reviewed case studies and city initiatives that included IoT to determine the current state of IoT-enabled smart cities. It covered important technology developments that helped to build a smart city. |
| 6 | 2022 | Bellini, Nesi & Pantaleo (2022) | The study categorized the main concepts and methods for smart cities into eight groups. It focused on vital technologies depicted in the literature for the implementation of IoT frameworks. |
| 7 | 2022 | Al-Turjman, Zahmatkesh & Shahroze (2022) | This study assessed how privacy and security protocols affected information-based smart city applications. It detected issues that must be fixed to boost output. |
| 8 | 2022 | Rejeb et al. (2022) | The article summarizes studies related to IoT and smart cities across the globe based on publications and bibliometric methods. |
| 9 | 2022 | Ahmad et al. (2022) | The study analysed in detail the main issues faced by smart cities. It lists the pitfalls and future paths of research in various areas and how it fills gaps and results in effective solutions. |
| 10 | 2020 | Chanal & Kakkasageri (2020) | The study examined obstacles to IoT regarding availability, confidentiality, integrity, and authentication. Comparing the trust models based on Function Requirements (FR) and Trust-Related Attacks (TRA) of IoT enabled researchers to assess each tier's security issues and solutions. |
| 11 | 2024 | Bauer, Sanchez & Song (2021) | Proposed a systematic literature review of 55 primary studies on IoT in smart cities, focusing on smart homes, infrastructure, and industry. Identified key security, privacy, and interoperability challenges and proposed a unified framework to enhance urban efficiency and sustainability. |
| 12 | 2024 | Nusir et al. (2024) | Experts identified several technologies, such as sustainable green infrastructure and advanced transportation systems, with significant potential for near-term success in Jordan's smart city development. |
| 13 | 2024 | Khalique et al. (2024) | The study provides an overview of the essential components of a smart city, such as IoT devices, communication networks, data analytics platforms, and service interfaces. |
| 14 | 2024 | Shah et al. (2024) | Research focuses on developing lightweight blockchain solutions and establishing standards for UAV operations within smart city frameworks. |

studies for smart cities. In addition, this study investigates the challenges of adopting IoT in smart cities and provides prevention methods for each challenge. Moreover, this study offers future directions for the upcoming researchers. It serves as a common ground that aids top-level decision-makers in making significant decisions on transforming cities into smart cities. The main contributions of this study are listed in the following:

- Comprehensively classifying IoT components in smart cities, including infrastructure, transportation, healthcare, waste management, and energy efficiency.
- Analyzing the role of AI, sensing technologies, and networking protocols in enhancing smart city functionalities.

- Conducting a comparative case study analysis of smart cities (*e.g.*, Masdar City, New York, Copenhagen, Singapore, Fujisawa) to highlight successful IoT implementations and best practices.
- Examining essential issues and future areas of study, with an eye towards cybersecurity concerns, big data management, interoperability, and policy frameworks to assist creative sustainable city development.

The rest of this article is organised as follows: Methodology presented in "Methodology". "The primary components of IoT-based Smart City" looks at the primary components of an IoT-based smart city. "Artificial Intelligence and IoT" explains how AI makes IoT systems smarter. "Sensing Technology for IoT-enabled Smart Cities" dives into the various sensors used for collecting data in cities. "Wireless Network and IoT Smart Cities Communication technologies" explores wireless network and communication technologies. "IoT Synergies and Case Studies" looks at IoT synergies and case studies. "Case studies Discussion" discusses the primary challenges to adopting IoT in smart cities and the problems cities might face in using IoT. "Primary Challenges for Adoption of IoT in Smart Cities" explores limitations and future research directions. Finally, "Conclusion" concludes the study findings.

## METHODOLOGY

This study employed a qualitative research design to comprehensively examine the role of the Internet of Things (IoT) in urban development, focusing on its application in smart cities (*Rai et al., 2023*). The research objectives listed in Table 2 below aimed to explore key components of IoT-based innovative city ecosystems, the integration of associated technologies, and the challenges faced. The methodology was structured into six phases, ensuring a detailed and systematic approach.

### Literature review and conceptual framework

The initial phase involved conducting an extensive literature review to identify the critical elements of IoT within the context of urban development. As outlined in Table 1, the review included a comprehensive examination of peer-reviewed journal articles, industry reports, and recent surveys focusing on the role of IoT in smart cities. Special attention was given to research on IoT applications in public transportation, urban infrastructure, and communication systems. This phase also explored various dimensions of smart cities, including the social impacts, economic scalability, and environmental sustainability of IoT in urban planning, particularly within developing regions. Additionally, the review incorporated prevalent IoT solutions designed for future urban environments, ensuring a holistic understanding of the global opportunities and challenges smart cities face.

By allowing the gathering, analysis, and use of enormous volumes of data from metropolitan surroundings, IoT provides a fundamental tool for urban development. IoT, according to *Hassan et al. (2020)*, allows daily objects from streetlights to vehicles into a cohesive network, enabling real-time monitoring and decision-making. Urban environments allow this capacity to be used to solve urgent problems such as

**Table 2 Research objectives.**

| Research objective | Motivation |
|---|---|
| 1. Assess the Role of IoT in Optimizing Smart City Infrastructure. | Explore how IoT technologies can enhance urban services such as transportation, energy management, waste disposal, and healthcare to improve efficiency and quality of life. |
| 2. Investigate Security and Privacy Challenges in IoT-based Smart Cities. | Analyze the key security risks and privacy concerns associated with IoT in smart cities, proposing innovative strategies to enhance data protection and trust. |
| 3. Examine the Integration of AI and IoT for Sustainable Urban Development. | Study the combined impact of AI and IoT in creating sustainable, energy-efficient cities, focusing on reducing environmental impact and improving resource management. |
| 4. Conduct Comparative Case Studies of IoT Adoption in Global Smart Cities. | Evaluate the implementation and success of IoT technologies across different smart cities worldwide, identifying key factors that contribute to their effectiveness and challenges. |
| 5. Analyze the Socioeconomic Impact of IoT in Urban Development. | Investigate the broader social and economic implications of IoT adoption in smart cities, including its effects on public engagement, job creation, and urban economic growth. |

environmental damage, energy inefficiencies, and traffic congestion. *Kaiser & Deb (2025)* contend that IoT is a pillar of smart city projects since its capacity to support sustainability and efficiency provides a structure for combining several urban systems into a single, responsive ecosystem.

IoT sensors and cameras track traffic patterns, control parking availability, and maximize public transit paths in the context of smart mobility (*Floris et al., 2022*). For example, smart traffic signals with IoT technology can change their timing depending on real-time traffic data, enhancing urban mobility and lowering congestion. In smart energy management, IoT sensors similarly provide real-time monitoring of building energy consumption, therefore enabling dynamic modifications that minimize waste and reduce costs (*G et al., 2024*). Smart trash management, where IoT-enabled bins notify when are full helps to maximize collecting schedules and lower running inefficiencies (*Sosunova & Porras, 2022*). These projects show how IoT uses data-driven solutions to address daily problems thereby improving urban functionality.

## Data collection

Data was collected from secondary sources, including existing IoT case studies and reports on innovative city implementations worldwide. Specific examples from cities like Masdar City in Abu Dhabi, New York, and Copenhagen were analyzed, providing diverse insights into how IoT technologies are integrated into urban infrastructure to address challenges such as urbanization, transportation, and sustainability to reflect different levels of urbanization and technological readiness.

## Analysis of IoT components

This phase focused on the technical aspects of IoT, including sensing technologies, wireless communication methods, and AI-based enhancements. The components were categorized

into innovative city domains: energy management, transportation systems, and waste management. The study highlighted emerging technologies like 5G networks, smart sensors, and edge computing, emphasizing their contributions to urban efficiency and sustainability. This ensures that the IoT systems are assessed across different domains, which inherently capture user interactions and experiences, especially in phases like "Analysis of IoT Components" and "Comparative Case Study Analysis.

## Comparative case study analysis

The study incorporated a comparative analysis of case studies from various cities that successfully implemented IoT-based solutions.

This study's selection of case studies was based on geographic diversity, technological maturity, and policy frameworks to ensure a comprehensive analysis of IoT implementation in smart cities.

The cities of Masdar City (UAE), New York (USA), Copenhagen (Denmark), Singapore, Fujisawa and Kashiwa-no-ha (Japan) were selected for their innovative IoT use cases and global applicability in the development of smart cities. Masdar City which ranks 28th as a smart city globally, is notable for its sustainability initiatives focused on renewable energy and autonomous vehicles (*Pandita et al., 2024*). New York has diverse uses of IoT, such as smart water meters and open data platforms with scalable urban solutions (*Hangan et al., 2022*). Copenhagen, with its vision of becoming CO2 neutral by 2025, operates IoT to control energy and track bike traffic, winning the 2021 World's Finest Smart City Initiative award (*Carpentiere, Messeni Petruzzelli & Ardito, 2024*). Singapore leads in IoT-driven applications like the "One Monitoring" transport system and smart garbage management, providing regional guidance (*Ferro-Escobar, Vacca-González & Gómez-Castillo, 2022*). Fujisawa and Kashiwa-no-ha lead the way in green-focused practices, ranging from solar energy sharing to high-tech waste recycling, exemplifying sustainable city development (*Barrett, DeWit & Yarime, 2021*). Together, these cities provide a range of representative challenges, innovations, and good practices for IoT-fueled smart city initiatives.

The cities from different regions, including Masdar City, New York, Copenhagen, Singapore, and Fujisawa, were chosen to capture variations in infrastructure readiness, economic conditions, and regulatory environments. Technological maturity played a crucial role (*Fantin Irudaya Raj & Appadurai, 2022*), as cities with well-established IoT infrastructures and advanced innovative city initiatives, such as Singapore and Copenhagen, were prioritized to highlight cutting-edge implementations. Additionally, policy frameworks and governance models were key factors in the selection process, with cities like New York and Masdar City included for their distinct regulatory approaches, cybersecurity measures, and public-private partnerships in innovative city development. By incorporating these criteria, this study ensures that its findings are applicable across different socioeconomic and technological landscapes, offering valuable insights into the global adoption of IoT in urban environments.

## Challenges and security considerations

A critical part of this methodology involved evaluating security challenges associated with IoT adoption in smart cities. This phase explored literature on cyberattacks, data privacy concerns, and the vulnerabilities of IoT systems. Strategies such as encryption methods, blockchain technologies, and regulatory frameworks were assessed to provide recommendations for enhancing security in IoT-based innovative city applications. For example, blockchain can secure IoT data by creating immutable transaction logs that prevent unauthorized alterations. This technology can be particularly effective in applications like smart healthcare and waste management, where data integrity is critical.

## Validation

The final phase focused on outlining validation methods to assess the real-time effectiveness of IoT components in urban environments. However, the discussed validation process did not involve using quantitative metrics such as user satisfaction, performance indicators from existing IoT systems, or user feedback surveys. Instead, the validation relied on a qualitative evaluation of IoT components, lacking concrete performance measurements or real-world pilot tests to substantiate the findings. Finally, while direct measurement of user satisfaction may not be present, the holistic approach adopted, spanning security, operational metrics, and real-world applications, offers a valid and sufficient framework for assessing the effectiveness of IoT systems in urban environments (*Bhardwaj et al., 2023*). For instance, improved transportation systems or competent healthcare can be inferred to enhance user experiences.

## THE PRIMARY COMPONENTS OF IOT-BASED SMART CITY

Smart cities are already in existence and are growing rapidly, the adoption of IoT has escalated due to its excellent service provision to intelligent city dwellers. Some benefits of using IoT are better power distribution, ease of garbage collection, smooth traffic flow, and improved air quality. Figure 2 depicts the primary component of IoT-based smart cities. The leading IoT components and applications in smart cities are discussed in the following subsections:

## Smart home

A smart home has technology that can communicate with its physical surroundings and non-physical objects. It gives its dwellers more control over their living space, better security, and more effective energy management. Many IoT options are available for house monitoring and development. In 2016, the market for smart home products was worth USD 5.65 billion, and by the end of 2025, it is expected to reach USD 174.24 billion, outpacing all prior predictions (*Yang, Liu & Gong, 2020*).

## Smart health

Healthcare devices are one of the most swiftly expanding sectors of the IoT market, better known as the Internet of Medical Things (IoMT), which has been projected to attain

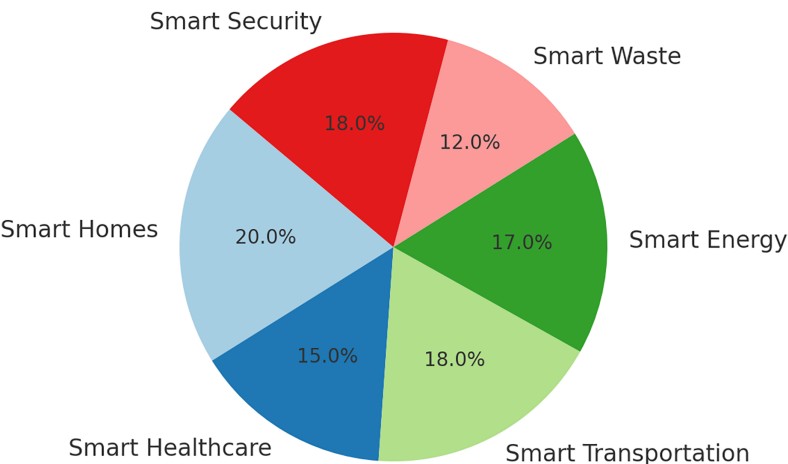

**Figure 2 The primary categories of sensing technologies in IoT-based smart cities.**

$176 billion by 2026 (*Razdan & Sharma, 2021*). IoT devices provide novel methods for healthcare providers to monitor patients and for people to track their health. The abundance of wearable IoT devices presents healthcare professionals and their patients with advantages and disadvantages.

**Monitoring of remote patient**

Monitoring remote patients has mainly involved using IoMT within the medical industry. Patients no longer need to travel to service providers since IoMT devices can collect health indicators from those not in a healthcare facility, including temperature, blood pressure, and heart rate (*Akkaş, Sokullu & Cetin, 2020*). An IoMT sensor that detects a patient's unusually low heart rate may alert medical personnel to intervene. A significant challenge of remote patient monitoring is maintaining the confidentiality and privacy of the highly sensitive data collected by these IoMT devices.

**Monitoring of glucose**

Conventional glucose monitoring has been proven difficult for a substantial portion of the world's diabetic population. By using IoMT, patients' glucose levels can be checked, verified, and documented manually. Routine testing might be inadequate to detect an issue, significantly when the glucose levels dramatically fluctuate (*Patel et al., 2023*). Hence, IoMT devices can facilitate resolving such a problem by continuously and automatically checking patients' blood sugar levels. With glucose monitors, manual record-keeping can be dismissed, and patients can be warned when their blood sugar levels exceed the standard rate.

**Heart rate monitoring**

The availability of numerous miniature IoMT devices that can monitor heart rate enables patients to live actively while having their heartbeat monitored continuously. Even

for individuals in the hospital setting, measuring cardiac rhythms can be challenging, like monitoring glucose levels. Fast swings in heart rate cannot be identified from routine heart rate checks, and patients must be continuously connected to wired devices when using the typical continuous heart monitoring techniques found in hospitals (*Patel et al., 2023*). It is still challenging to achieve ultra-accurate results, although most modern machines can attain accuracy rates up to 90% or higher. Imminently, the use of IoMT ensures a healthier heart.

### Monitoring of depression and mood

Information about patients' depressive symptoms and general mood is often challenging to gather regularly. Healthcare professionals cannot foresee sudden mood swings despite routinely asking patients how feel. Patients usually fail to express their feelings appropriately. Therefore, IoMT devices that can detect moods may address this problem. By gathering and analyzing information, such as heart rate and blood pressure, the devices can deduce details about one's mental state (*Gupta, Joshi & Kulkarni, 2022*). Modern IoMT devices for mood monitoring can track data, such as eye movement. One issue noted here is that assessments of this nature cannot reliably predict anxiety or other related symptoms, let alone depression symptoms.

### Inhalers

Asthma and chronic obstructive lung disease are two examples of conditions that frequently cause unexpected episodes. The IoMT-connected inhalers can aid patients by monitoring seizure frequency and gathering environmental data to assist healthcare practitioners in identifying the cause of seizures (*Pradeesh et al., 2022*). When patients forget their inhalers at home and are in danger of having a seizure or using their inhalers improperly, linked inhalers can alert them.

### Contact lenses

Another innovative and unobtrusive method of capturing health data is using smart contact lenses. These lenses have microscopic cameras that enable users to capture images with their eyes (*Xiang et al., 2022*), which is why some companies (*e.g.*, Google) have trademarked IoT contact lenses. Smart glasses can turn the human eye into a potent tool for digital interactions, whether for improving health outcomes or for numerous other purposes.

### Robotic surgery

Surgeons can now do complex procedures that would be challenging for human hands by utilizing small, internet-connected robots within the human body. Robotic procedures employing miniature IoMT devices can reduce the size of surgical incisions, resulting in a less invasive approach and expedited recovery for patients (*Dwivedi, Mehrotra & Chandra, 2022*). For surgeons to perform surgery with the least amount of disruption and difficulty possible, the devices must be compact and trustworthy. To make wise decisions during surgery, one must understand the complex body system. The use of IoMT robots in surgery denotes that some issues can be effectively resolved.

## Smart industry

In the manufacturing industry, the Industrial Internet of Things (IIOT) is the industrial subset of the IoT. Notably, IIoT facilitates the realization of the fourth industrial revolution (IR 4.0). Businesses have begun using IoT-connected manufacturing solutions, such as Azure IoT, which manages IIoT devices. Cloud software linked to various resources can be mobilized to control devices. Key metric data (*i.e.*, equipment efficiency and telemetry data) can be reported *via* IIoT, and data can be retrieved from assets located in distinct geographic locations. Connected factory solutions may be applied to connect, monitor, and operate industrial devices from a distance (*Zakari et al., 2022*).

## Smart energy

The IoT is used as part of a complete energy management approach called "smart energy" to distribute energy in a cost-efficient and efficiency-conscious manner. It entails using sustainable and renewable energy sources with IoT power management systems and equipment (*Jha et al., 2024*). The IoT-enabled smart energy and utility solutions link sustainable energy assets. These IoT-enabled devices are integrated into IT infrastructure to increase energy efficiency and factory-to-customer delivery. The main applications of IoT smart energy are elaborated in the following:

### Oil and gas

Intel and ExxonMobil, for example, collaborated to develop a Universal Wellpad Controller (UWC) to help oil and gas companies manage and grow their field automation more quickly and cost-effectively. Another example is the Universal Wellpad Controller (UWC), which was created in partnership with ExxonMobil and is a powerful edge control device for the oil and gas sector. It automates the monitoring and control of onshore production wells and fields, increasing flexibility and lowering costs. Many operations in the upstream, intermediate, and downstream segments have integrated IoT-enabled solutions due to their significant impact and socioeconomic benefits (*e.g.*, condition-based monitoring and location tracking). However, some barriers (*e.g.*, susceptibility to cyberattacks, poor communications networks, lack of technological readiness for usage in zone-0 and zone-1 hazardous environments, labour concerns, as well as maintenance and obsolescence) have slowed the adoption of IoT devices for numerous operational processes (*Wanasinghe et al., 2020*).

### Wind turbines

The new energy business has boosted its use by integrating the latest technologies, such as IoT, AI, and cloud computing. Smart approaches not only address issues but have emerged as the preferred option for new energy firms looking to take the lead in future green energy growth. Wind and photovoltaics (PV) are two examples of clean, renewable energy that are gaining popularity. As environmental elements (*i.e.*, wind direction, wind speed, temperature, sunshine, and air pressure) have a significant impact on power systems during the power generating process (*Karad & Thakur, 2021*), the equipment and grids tend to experience operational and safety issues. Due to the increasing adoption of clean energy concepts, renewable energies (*e.g.*, wind power and PV) have a crucial role in the energy industry. Based on information obtained from China's National Energy

Administration (NEA) (*Xu et al., 2020*), wind power output for the first half of 2019 showed a significant growth of 11.5% higher than the prior year, while 20.0% with installed PV capacity.

**Microgrids**

Energy users can participate actively in the energy sector by using microgrids - a network of electric circuits that enable two-way energy flow with the potential to self-heal when maintenance issues arise (*Routray et al., 2021*). Consider a scenario in which a neighbourhood of ten homes is transformed into a microgrid, and each home is connected. As a result, these homes can self-generate electricity for their own needs due to the solar panels installed on their roofs. However, when one residence produces more energy than it would use, any excess energy is wasted. When microgrids are used, energy is sent to the grids so that other entities can use them for their needs. The IoT can disconnect energy circuits from homes that do not wish to distribute energy.

**Smart meter technology**

Smart meter technology enables a household, building or organization to monitor and evaluate energy usage. These IoT applications in smart energy can be employed in the energy sector. One can examine the places where energy is used the most and the areas where energy can be saved with smart meter installation technology (*Routray et al., 2021*). Smart meters with IoT capabilities may be installed in residences and commercial buildings to provide consumers with a complete report on their energy usage trends. With the ability to detect vulnerabilities, smart meter technology can improve both the transparency and efficiency of the energy sector. Besides, smart meters enable two-way communication between utility companies and consumers, thus linking both parties instantly.

## Smart infrastructure

Any modification to structural characteristics that could jeopardize safety or raise risk can be monitored using the IoT infrastructure. A key IoT application is the monitoring and management of the operations of urban and rural infrastructures, such as railway lines and bridges, as well as on- and offshore wind farms. The IoT can be used to plan maintenance and repair work by efficiently coordinating duties between different service providers and users of those facilities. IoT devices, such as bridges that allow ships to dock, can manage critical infrastructures efficiently. The adoption of IoT devices for infrastructure monitoring and management enhances the coordination of emergency response efforts, incident management, and quality of service across all infrastructure-related fields (*Mehmood, Katib & Chlamtac, 2020*). The IoT smart infrastructure's automation and improvement can facilitate wastewater management.

## Smart city services

Embedded sensors that have track site-specific conditions, including frozen bridge surfaces, can assist with traffic rerouting and deploy fleets of snow blowers and salt trucks. Such improvements are more straightforward to justify, especially when faced with funding constraints because lives are at risk when road maintenance is neglected. Future

smart cities will depend solely on nationwide network devices accessible to everyone. In future, all connected devices will be necessary for smart cities, with most of them being paired with driverless vehicles. Smart buildings, smart transportation, smart parking, and smart lighting are some of the major IoT applications that will be implemented in smart cities.

### Intelligent automated transportation

Smart city automobiles can immediately communicate their position and status to the authorities *via* public networks, besides alerting other smart cars to reduce their speed. In preventing accidents, some smart transportation technology goes even further. IoT technologies can facilitate the coordination of communications, management, and information processing among multiple transportation networks. Numerous sectors of the transportation system have commenced the adoption of IoT, including inter- and intra-vehicle communication, intelligent parking, electronic toll collection systems, intelligent traffic management, logistics and fleet administration, vehicle control, safety and roadside assistance, all facilitated by the dynamic interaction of diverse components within the transportation system. New ways of public transportation are bound to emerge, such as a smart bus for the public with on-demand access and forthcoming technologies such as drone (flying) taxis (*Oladimeji et al., 2023*). The most important approach to gain access to these smart technologies is *via* network connectivity.

### Smart parking

Smart cars are used in some parking lots to communicate with each other and gather data on the number of available spaces. Vehicle input is used to locate specific gaps and auto-park. Street parking technology is beneficial in terms of gradually removing vehicles from congested areas and indicating particular parking zones at certain hours (*Channamallu et al., 2023*). Due to the limited available space, drivers waste time searching for parking spots, which can cause road congestion. This issue can be resolved by installing IoT-connected sensors throughout the city that indicate empty parking spots. This information helps municipal officials to determine congestion due to limited parking space and areas with plenty of open space. This technology optimizes parking while hindering traffic bottlenecks and driver annoyance.

### Smart building

Erecting smart buildings that prioritize the residents has received a fresh focus after the outbreak of Coronavirus Disease 2019 (COVID-19) (*Megahed & Abdel-Kader, 2022*). A central network that serves as the network "edge" is linked to the IoT and is the foundation of smart buildings. The IoT devices installed in smart buildings function as sensors that collect data and transfer it securely to the central network, frequently installed in remote or difficult-to-reach regions. Also, serve as an automated systems for regulating the lighting and window coverings according to the surroundings. The IoT devices may be connected to conference room equipment and office furniture to enhance their flexibility, which, in turn, can increase workplace efficiency. These devices consist of security-related gadgets, such as badge readers, remote cameras, and electronic door locks. For example, Indoor Air Quality Status (IAQ), invented by PHILIPS, uses mobile tools for indoor air quality

monitoring. The IAQ state includes a gateway, up to six IAQ sensors, and a browser-based FeelPlace Classic user interface. The IAQ sensors measure temperature, carbon dioxide, relative humidity, and volatile organic compounds (VOCs).

**Smart lighting or traffic light**

A segment of the local, wireless, distributed platform with local or cloud-based intelligence refers to the smart, linked lighting system. Data are received *via* sensors installed on lampposts, such as cameras, sunlight, movement, or noise detection, and processed to generate optimal energy-efficient and safety-supporting public lighting operations. Currently, cities across the EU are spending 20% or more of their energy on lighting, which is over 25 years of inefficient lighting, with 75% on public lighting assets (Program 2021). Many cities across Asia, Europe, and North America have already implemented IoT-based smart lighting technology by transitioning to energy-efficient Light Emitting Diodes (LEDs), with some even progressing to smart lighting with dimming and safety-supporting controls. For instance, Kansas City, Missouri, deploys bright public lighting, including streetlights with WiFi sensors. The system contributes to a live city map that informs the residents of the precise locations of public transportation throughout the city, speed of traffic, and empty parking lots.

## Smart agriculture

Smart agriculture linked to the IoT is meant to improve the agriculture sector *via* efficient data collection and analysis, innovative storage, excellent distribution of crops, and effectively overcoming many problems farmers face. The implementation of IoT for smart agriculture offers automatic irrigation by using a sensitive sensor to monitor plant water consumption and calculate the amount of water flow to plants for good crop cultivation. It provides soil health monitoring through special monitoring sensors to determine the health of crops, soil composition, and moisture level. In addition, integrated IoT devices in smart agriculture provide weather forecasting and knowledge of weather conditions based on temperature and cold sensors (*Aldossary, Alharbi & Hassan, 2024*).

Smart agriculture provides an effective way to control weeds that can harm crops. The quality of agricultural products can be improved through analysis, knowledge, and data collection for better agricultural planning. Agriculturists can utilize IoT-enabled smart farming applications to enhance various tasks, including identifying optimal harvest periods, formulating fertilizer profiles based on soil chemistry, collecting data on temperature, precipitation, humidity, wind velocity, and pest prevalence, as well as measuring soil nutrient and moisture levels. Smart Elements refers to one example of an IoT gadget used in agriculture, along with AllMETOE and Pynco (*Mahbub, 2020*). These sensors can track environmental information about plants and the weather. In August 2018, Toyota Tsusho and Microsoft announced a collaboration to create aquaculture solutions utilizing the Microsoft Azure application suite for IoT technologies pertaining to water management (*Pranto et al., 2021*). The water pump mechanisms utilize AI to quantify, assess, and compute the number of fish based on data obtained from the fish. Researchers from Kindai University helped create them in part. The FarmBeats project

from Microsoft Research links farms using TV white space and is accessible on the Azure Marketplace (*Ojha, Misra & Raghuwanshi, 2021*).

### Smart waste

Garbage management refers to keeping rubbish from spreading from its origin until it is disposed of *Kumar (2024)*. Some stages in this process include classifying rubbish, *i.e.*, if it is biodegradable, and locating a suitable disposal method. Even waste is classified as wet or dry. Hence, IoT-enabled devices can be used to recognize the different types of rubbish. These IoT devices ease laborious and mistake-prone tasks. For example, leaders in Chicago have linked data from garbage with sensors to instruments for predictive analytics. By training to anticipate the overflow of garbage cans and using that knowledge to reevaluate pest management tactics, the city has been 20% more effective at reducing the rat population. IBM has also created methods for recording particular events, such as dump trucks lifting rubbish without spills and new locations for landfills. Smart city IoT systems use these data to streamline collection processes, reduce emissions from moving vehicles, and enhance service quality.

## ARTIFICIAL INTELLIGENCE AND IOT

One of the most essential characteristics of smart cities is the application of AI in municipal administration. A smart city, which reflects the digital revolution, creates a massive quantity of data through the sensors dispersed across the city; the data are useless until analyzed and relevant information is retrieved (*Idhalama & Oredo, 2024*). As such, the role of AI in this case is important for digesting this massive quantity of data, as well as extracting facts and patterns that can facilitate the local government in taking appropriate actions. In the case of a power surge, for example, AI has the ability to learn the source of the surge and its related conditions. Such knowledge can be utilized to manage the power grid better. In addition, smart cities may benefit from AI when hosting significant events by using video cameras connected to AI-based analytics technologies. The AI algorithms can be used to examine the footage for behavioural or environmental irregularities to hinder any act of terrorism or violence from occurring.

AI techniques can be deployed to handle arising challenges in a two-stage procedure. In the initial step of the process, a collection of AI models is generated by employing machine learning techniques and training datasets; the best models are created by using a massive quantity of training data. After that, these new models may be used to conclude sensor input data and direct system operation (*Anthony Jnr, 2024*).

Smart cities use AI and IoT technologies to improve urban life, increase efficiency, and address several difficulties. Some specific AI incorporated with IoT technologies widely employed in smart cities may include:

**Smart traffic control:** (AI) algorithms analyze real-time data from sensors and other sources to improve traffic flow and eliminate congestion. Barcelona has deployed the Smart Mobility Barcelona system for automated traffic control (*Masood et al., 2023*; *Ouallane et al., 2022*; *Uddin et al., 2021*).

**Energy management:** AI algorithms control buildings' energy usage, lighting, and other infrastructure in response to real-time data and demand changes (*Singh & Ahmed, 2021*).

**Environmental monitoring:** AI analyzes information gathered from sensors, satellites, and meteorological stations to track air quality, disturbances, disposal of waste, and the condition of water (*Krishnan et al., 2022*).

**Waste management**: AI technologies, particularly the ones for sorting and treating solid waste, optimize garbage processing and recycling, reducing human interaction while increasing accuracy and speed. AI technologies, particularly the ones for sorting and treating solid waste, transform non-recyclable garbage into reusable goods, energy, and fuel. AI can enhance waste management, such as visual perception, machine learning, robots, and smart sensors (*Bharti, Fatma & Kumar, 2022*; *Fidowaty, Wulantika & Mulyana, 2022*; *Nasreen Banu & Metilda Florence, 2021*).

**Public safety and surveillance:** AI-powered facial recognition technology is rapidly being employed in public safety and surveillance applications. This technology uses AI algorithms to identify criminal suspects, find missing people, and detect possible threats by using biometrics to map facial traits from a photo or video. This is then compared to a database of known faces to determine a match. This technology is employed in smart cities mainly to resolve crimes and identify missing individuals (*Doohan et al., 2022*; *Mahor et al., 2023*).

**Smart water management:** Machine Learning for Conservation, an artificial intelligence system that evaluates use trends to optimize water consumption and uncover conservation opportunities (*Krishnan et al., 2022*; *Singh & Ahmed, 2021*).

**Smart building management:** AI automation systems that employ IoT sensors to monitor and regulate building factors, including lighting, temperature, and occupancy, as well as AI predictive maintenance that facilitates algorithms to evaluate data from building systems to forecast and schedule maintenance work (*Zhang et al., 2021*).

## SENSING TECHNOLOGY FOR IOT-ENABLED SMART CITIES

Smart devices and sensors are cutting-edge techniques that have transformed urban public services. Sensor networks refer to IoT enablers. Smart sensors are used for different applications, such as to improve traffic management, waste disposal, health, quality of air, as well as tracking of food and water (*Alahi et al., 2023*). These are some of the basic criteria for a functioning city. Schedules in conventional systems control these solutions. The significance of sensor technologies in smart cities can expand people's sensitivities and make cities bearable (*Hussain, 2024*). A smart city is impossible without sensor placement. A sensor in a smart city is used to measure the physical attributes of any object or scenario. Electronic sensors, infrared (IR) sensors, thermal sensors, proximity sensors, and biosensors are the most popular kinds of sensors that are installed. The functions of these sensors are discussed in the subsequent sections and Fig. 3 shows the primary categories of sensing technologies in IoT based smart cities.

### Electronic sensors

Environmental surveillance and speedometer sensors, prevalent in smart cities, utilize electronic sensors to execute several functions, including monitoring power and current

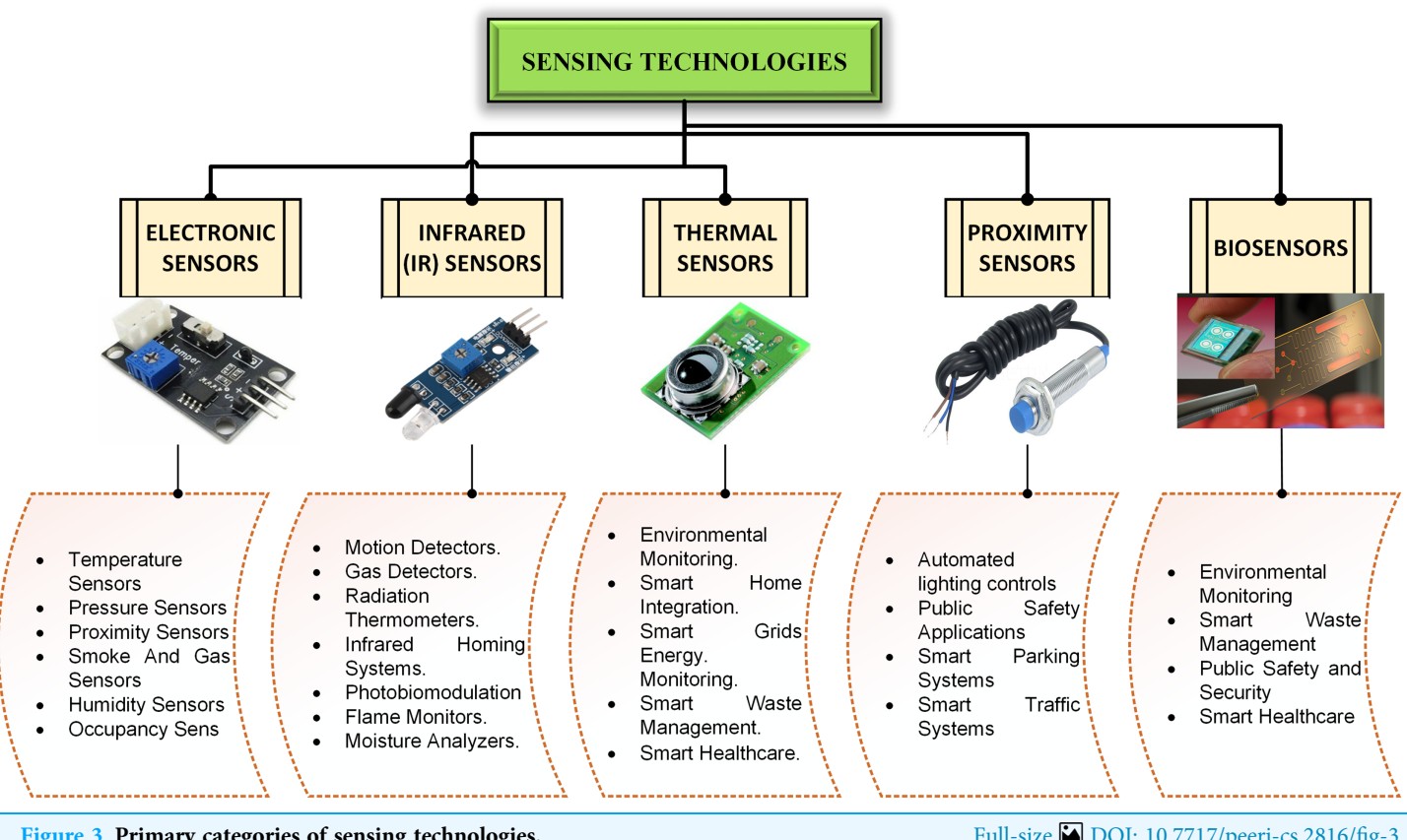

**Figure 3** Primary categories of sensing technologies.

levels to detect issues (*Omrany et al., 2024*). The primary role of electronic sensors is to detect various forms of energy *via* electroscopes, magnetic anomalies, and voltage detectors, among others. Environmental surveillance sensors, parking sensors, and speedometer sensors exemplify electrical sensors. Nonetheless, these wireless sensors exhibit multiple disadvantages, such as elevated energy consumption, inadequate flexibility, and significant complexity. Monitoring power and current levels facilitates the prompt detection of mistakes, ensuring the effectiveness of electronic signal transmission.

### Infrared sensors

Infrared (IR) sensors, which are electronic devices that monitor and detect IR radiation in their surroundings, can facilitate the decision-making process in a smart city. In erratic and dynamic surroundings, these sensors assist in collecting objective data. These sensors can be divided into active and passive IR sensors. An active IR sensor has two components: an LED and a receiver (*Akhtar, Rahman & Lazoglu, 2024*). As objects approach the sensor, the IR light from the LED reflects off them, thus enabling the receiver to detect them. Obstacle detection systems often use active IR sensors as proximity sensors (*e.g.*, for robots). The most popular use for passive IR (PIR) sensors, which are electronic sensors

that capture IR light produced by objects in their field of view, is PIR-based motion detectors. PIR sensors are frequently used in automated lighting and security alarm systems. Without identifying a moving individual or thing, these sensors can detect the overall movement (*Akhtar, Rahman & Lazoglu, 2024*).

**Thermal sensors**

A thermal sensor measures the temperature of a system or the environment (*Tomat, Ramallo-González & Skarmeta Gómez, 2020*). These sensors are primarily designed for various applications varying from industrial process monitoring to environmental management. This type of sensor can detect thermal signals, such as temperature, acceleration, angular velocity, and flow. Thermocouples are the most often used heat sensors in automotive, consumer, and industrial applications.

**Proximity sensors**

Proximity and LIDAR sensors assist in the development of autonomous vehicle systems, which is crucial for turning a city into a smart city. Proximity sensors detect items nearby without using touch. IR radiation beams or electromagnetic fields are frequently emitted by these sensors. There are many fascinating uses for proximity sensors, such as picking up movement in retail between a customer and a potential purchase. If there are product sales or promotional offers close to the sensor, the user will be informed. At malls, stadiums, and airports, proximity sensors are frequently applied to display parking capacity. Also, be used in assembly lines for various industries, including food and chemical industries (*Hernández-Heredia et al., 2024*).

**Biosensors**

A biosensor denotes a biological sensor. The device comprises a transducer and a biological component, such as an enzyme, antibody, or nucleic acid. The transducer turns the biological reaction into an electrical signal after the biological element and the analyte under investigation come into contact. All biosensors have two parts: a biological sensor and an electronic component that receives and sends signals. When the sensor interacts with the analyte, it generates a signal that might be electrical, optical or thermal. Next, it is transformed into a quantifiable electrical parameter, generally a current or voltage, by using an appropriate transducer. The role of biosensors in biomedicine is to detect analytes. Advanced biosensor technologies include ionizing and subatomic sensors, such as neutron and MEMS sensors. Nonetheless, biosensors are limited to the domain of biomedicine (*Mishra et al., 2024*).

## WIRELESS NETWORK AND IOT SMART CITIES COMMUNICATION TECHNOLOGIES

Wireless Personal Area Network (WPAN), Wireless Local Area Network (WLAN), and Wireless Metropolitan Area Network (WMAN) constitute the three classifications of wireless technologies utilized in smart cities. Field Area Network (FAN) and Body Area Network (BAN) are now represented by new terminologies such as Home Area Network (HAN) and Neighborhood Area Network (NAN). As depicted in

*Kareemulla & Khasim (2024)*, smart city communication technologies can be grouped into three groups (*Kareemulla & Khasim, 2024*):

### Cellular mobile networks

To connect gadgets with the Internet, cellular technology is a standard option. As IoT devices require connection to communicate with gateways, applications, servers, and cloud platforms, cellular IoT wirelessly links them to the Internet. Cellular networks cover almost every corner of the globe, and IoT has piggybacked on that existing infrastructure. The necessity for global cellular connectivity has encouraged the establishment of global standards, thus implying universal technology alignment. Although standards (*e.g.*, 2G, 3G, 4G, 5G, NB-IoT, and LTE-M) may seem a complicated jumble of jargon, international organizations (*i.e.*, 3rd Generation Partnership Project (3GPP) and GSM Association (GSMA)) keep everything under control. Other networking technologies either miss out on the benefits of global standards or do not have the capacity to interface (*e.g.*, SigFox, LoRa, Bluetooth, and WiFi).

### IoT-dedicated cellular networks

A private cellular network refers to an on-premises network for an organization's sole usage and requirements. A private (mobile) network is one in which network infrastructure is only used by devices authorized by the end-user company.

This infrastructure is often implemented in one or more sites owned or inhabited by end-user companies. Devices registered on public mobile networks will not function on private networks unless expressly approved. Although these are formally classified as "non-public networks", "private network" is generally used across vertical businesses. These cellular networks are available in 4G and 5G configurations. The advent of the 5G network and its capabilities enhanced the delivery of private and dedicated networks, thus building on the accomplishments of organizations that implemented such networks using 4G LTE.

### Multi-tier architectures networks

Various IoT applications demand varying degrees of intelligence and efficiency in data processing. In offering future IoT services, multi-tier architectures incorporating cloud, fog, and edge computing technologies will be necessary. Multi-tier architectures cover IEEE 802.15.4, ZigBee, WI-SUN (802.15.4 g), ULP (802.15.4q), Wireless M-Bus, Z-Wave, Bluetooth Low Energy (BLE), WiFi Low Power (802.11ah), and other technologies (*Mohamed & Hamad, 2020*). Such technologies offer flexible configuration and simple reconfiguration, which are intended for applications with a few nodes and limited communication range. Table 3 compares the network technologies for IoT smart cities.

## IOT SYNERGIES AND CASE STUDIES

The following case studies examine diverse smart city efforts globally, emphasizing the role of IoT technologies in fostering advances in urban development, transportation, energy management, and sustainability. Each case illustrates the distinct methods by which cities utilize IoT to elevate inhabitants' quality of life, augment operational efficiency, and

**Table 3 Network technology comparison for IoT smart cities.**

| Communication technology | Technologic solution | Use case | Advantages | Disadvantages | Core network | Applications |
|---|---|---|---|---|---|---|
| Cellular mobile networks | - 2G Networks<br>- 3G Networks<br>- LTE<br>- 4G Networks<br>- 5G Networks | Smart Home | - Global<br>- Standardized | - When you travel, your home network is unaware of what you are doing in actual environment<br>- Consume more power to be a viable option for IoT devices<br>- The licensed spectrum is costly to acquire and use. | - Home Location Register (HLR)<br>- Home Subscriber Server (HSS)<br>- Packet data network GateWay (PGW) | - Telematics and connected cars<br>- Automated meter reading<br>- Wearable devices<br>- Smart waste management |
| IoT-dedicated cellular networks | - 4G Networks<br>- 5G Networks | Industrial IoT (IIoT) | - Support reliable high-bandwidth wireless connectivity and capacity<br>- Local ultra-low latency network infrastructure<br>- High bandwidth transmission of video imagery and location positioning<br>- Lower costs for network equipment and devices | IoT-specific systems frequently have constrained downlink channels and an unequal supported channel rate at the air interface | - LTE-M<br>- NB-IoT | - Driver safety assistance services Vehicle to Vehicle & Vehicle to Infrastructure<br>- Drone to drone communications for collision avoidance<br>- Access control/ escape route monitoring & control |
| Multi-Tier architectures networks | - Solutions Based on IEEE 802.15.4<br>- ZigBee<br>- 6LoWPAN | Indoor scenario (*e.g.*, home automation or industrial control/ metering) | - Extremely adaptable configurations<br>- Extremely customizable based on customer needs | Typically required to make up for the short communication range provided by the prevailing radio technologies | - Wireless short- and medium-range technologies<br>- Technologies for long-distance backhauling communications | - Smart security cameras<br>- Automatic heating and cooling<br>- Automated material handling systems |

advance environmental objectives—additionally, employing IoT technologies to oversee transportation, electricity, garbage, and other essential infrastructures, fostering smarter, more sustainable urban settings. This example outlines smart city initiatives in Masdar City (Abu Dhabi, UAE), New York (USA), Copenhagen (Denmark), Singapore, and the Japanese cities of Fujisawa and Kashiwa-no-ha.

**Masdar City in Abu Dhabi–UAE**

Abu Dhabi and Dubai topped the first and second places regarding their status as smart cities. Dubai has been named the smartest city in the Middle East and North Africa for 2022. As stated by *Bellini, Nesi & Pantaleo (2022)*, Abu Dhabi was ranked 28th in the

world, and Dubai was right behind it in 29th out of 118 cities. In comparison to 2021, the two cities rose 14 spots. Since 2006, the Emirate of Abu Dhabi has started preparing for Masdar City. The following are the primary aspects of the Masdar City master plan:

- The most effective use of solar energy through the creation of efficiency improvements and the best placement of the city's electricity infrastructure
- The integration of all parts of city life in a way that makes life simple and enjoyable while housing that is accessible nearby reduces the need for transportation
- Structures and buildings with restricted heights and dimensions
- Creating and expanding public spaces for a sociable and contented existence while giving pedestrians and a variety of transit options consideration
- Ensuring a high standard of living within an intelligent ecology

The primary smart city service pertains to its intelligent transportation system, exemplified by the 'Autonomous vehicle produced by the prominent French firm 'Navia'. It is an autonomous vehicle designed for first and last-mile transportation, facilitating the movement of passengers to and from transit terminals. It features eight seats that can accommodate 12 passengers, with a maximum speed of 25 km/h. The vehicle is outfitted with LIDAR cameras and sensors that generate 2D and 3D maps for obstacle detection. The global positioning system (GPS) is implemented to ascertain the vehicle's location, while the V2X system facilitates communication with traffic signals. A self-driving vehicle service was just initiated to operate consistently throughout the day, linking various car parks to the city's major centre. The inaugural autonomous vehicle transportation service available in the Middle East and North Africa region (*Sankaran & Chopra, 2020*).

### New York–USA

New York City has numerous IoT projects, as listed in the following (*Cirillo et al., 2020*):

- Open data for all: Some regional organizations make free public data available. With this tool, anyone can use data to improve their communities, including teachers, students, artists, builders, small business owners, campaigners, journalists, and community board members. It implies that open data for 300,000 employees contributes to a cleaner, safer, and more egalitarian New York City.
- LinkNYC offers free and fast WiFi, calls, device charging, and a tablet with access to municipal services, maps, and directions. Due to this improved communications network, payphones are no longer used in the Bronx, Brooklyn, Manhattan, Queens, and Staten Island.
- MyNYCHA: A smartphone application and web interface enable public housing residents to administer services online. In addition, can pay rent, examine inspection schedules, and sign up for project notifications.
- Automated water meters in NYC: The individual water meters are each connected to a small device as part of an automated meter reading system. Daily readings are transmitted to a computerized invoicing system.

- My DEP account allows New Yorkers to monitor their consumption remotely and dismisses the need for a water meter reader to visit the property. It also enables the Department of Environmental Protection to manage the city's water supply system better and monitor water use across the board.

### Copenhagen–Denmark

In Denmark, Copenhagen won the world's finest Smart City initiative award in 2021. The mayor of technical and environmental affairs in Copenhagen received the award. The top IoT-based smart city of Copenhagen has the following aspects (*Bundgaard & Borrás, 2021*):

- City's bike traffic: Copenhagen uses sensors to track bike traffic in real time, which offers useful information to enhance its bike routes. Given that daily bike commuting accounts for more than 40% of the city's residents, this idea is incredibly significant.
- A house of energy—with cooling and heating: The cooling system in Copenhagen largely uses seawater to cool the water supplied as year-round cooling for servers and air conditioning in business buildings (*e.g.*, hotels and banks). The HOFOR has set up several heat pumps across Copenhagen to draw energy from groundwater, sewage water, and even the ocean. The measures put in place by HOFOR in Nordhavn assist Copenhagen residents in reducing their carbon emissions and helping the city reach its ambitious goal of becoming CO2-neutral by 2025. These ideas can be scaled to new metropolitan regions in significant cities worldwide after being tried and true in Nordhavn. Some of the primary IoT-based energy sources can provide the capital with even greener district heating.

### Singapore

Singapore is a successful example of a smart city in terms of its water network (*Shamsuzzoha et al., 2021*):

- Leak management and network operations enhancement: It uses acoustic sensors *via* water flow acceleration and hydrophone sensors to capture sound waves underwater. Pressure sensors record pressure and flow data with the National University of Singapore (NUS) (*Chang & Das, 2020*).
- 'One Monitoring' transportation system: A team from Nanyang Technological University (NTU) developed an IoT transportation system called 'One Monitoring' to boost the efficacy of Singapore's public transportation. The system uses sensors to track the locations of buses and trains and the number of passengers on board. The data are used to improve routes and schedules for the public transportation network in Singapore.
- Parking guidance system: A parking guidance system that informs drivers of parking availability in real time.
- Smart waste management system: Smart bins have been used as part of the smart waste management system since 2015.

**Fujisawa and Kashiwa-no-ha-Japan**

Smart cities typically practice environmental protection, with structures that produce clean energy and heat efficiently. The Japanese smart city is a nice illustration of this (*Gornik, 2020*):

- Fujisawa Smart City: Built in 2010 on a former Panasonic factory site, it holds about 100 innovative residences, each fitted with solar panels and natural gas-powered generators spread across 190,000 square meters. The generated energy is automatically shared across all the households because connected through a single network. Panasonic has designed the city based on total reliance on electric power and the reduction of environmental pollution emitted from cars that run on regular fuel. Hence, it has installed power stations across the city. If this city is cut off from external energy sources, it can suffice for three full days of energy.
- E-waste recycling: The municipality of Fujisawa uses IoT to automatically separate various wastes utilizing a range of sensors, such as IR and ultrasonic sensors. The operation of an E-waste recycling system starts with the collection of garbage. A sensor is installed in the garbage bin to detect wet and dry wastes. The sensor is linked to certain interior bins to sort the waste effectively. Workers do not need to monitor the system manually but use electronic devices that protect them from harmful chemicals and odours.

HITECH developed the Kashiwa-no-ha Energy Management System (AEMS) to achieve oversight, efficiency, and control of local energy usage. The system shares electricity across provincial borders, reduces CO2 emissions, and provides energy data. During an emergency, electricity is distributed so that priority is given to elevators, evacuation centres, and other vital functions.

## CASE STUDIES DISCUSSION

Most current research such as *Kaiser (2024)* indicates that smart cities flourish when match technological innovation with local challenges. While Fujisawa shows the possibilities of designed resilience, New York's strength resides in retrofitting a vast metropolitan infrastructure. Given their scalable, pragmatic solutions, Copenhagen and Singapore seem the most flexible; Masdar and Fujisawa shine in regulated conditions. In the end, the success of a smart city depends on striking a balance between ambition and practicality, a lesson clear from many different models worldwide.

Using its Navia technology, Masdar City pioneer's autonomous vehicles with an eye toward first- and last-mile connection. Although (2020) (*Linke*) acknowledges this as a regional milestone, its low speed (25 km/h) would limit more general acceptance. Copenhagen, on the other hand, shines in sustainable mobility by using real-time bike traffic sensors to complement its 40% riding commuter base, therefore proving an efficient but low-tech method fit for its urban setting. Using IoT, Singapore's "One Monitoring" system improves public transportation efficiency using a data-driven alternative to Masdar's vehicle-centric approach, which *Zhou et al. (2023)* propose could become a model for other crowded metropolitan centres. Though it lacks the sophisticated transit

technologies seen in other cities, New York replaces out-of-date payphones with LinkNYC's WiFi and communication hubs, thus improving connectivity.

With limited building heights and integrated planning to reduce energy waste, Masdar City gives solar energy efficiency top priority together with low-carbon urban architecture, then complementing the sustainability objectives of the UAE. Aiming toward carbon neutrality by 2025, Copenhagen employs a more all-encompassing approach combining seawater cooling and heat pumps to lower CO2 emissions (*Terenius, Garraghan & Harper, 2023*). A networked energy system combining solar panels and gas generators guarantees self-sufficiency for up to 3 days. On the other hand, stressing sustainability within current metropolitan systems, New York and Singapore concentrate on resource management including automated water meters and leak detection.

IoT deployment in smart cities presents key difficulties even with its potential. Technical challenges include interoperability among diverse devices, data security, and the intense cost of infrastructure changes are noted by *Albouq et al. (2022)*. Masdar City's reliance on advanced sensors and autonomous systems, for example, calls for significant investment and could thus restrict its replicability in less wealthy areas (*Block, Hansen & Steinmetz, 2023*). As shown by New York's open data projects, where massive data collecting begs for ethical considerations regarding surveillance and consent, privacy issues also loom big (*Sideri et al., 2022*). Likewise, although efficient, Copenhagen's dense sensor networks call for strong cybersecurity protocols to stop intrusions (*Zang, 2023*).

# PRIMARY CHALLENGES FOR ADOPTION OF IOT IN SMART CITIES

The execution, governance, management, and level of collaboration among residents, businesses, and local governments of smart city projects are all exceedingly intricate and fraught with difficulties. Many businesses resist the adoption of IoT due to uncertainties in terms of compatibility, security, and privacy. IoT security revolves around maintaining the integrity of applications running on devices, enabling device and user identification, clear ownership of devices (and the data created by the devices), and being resilient to cyber and physical threats (*Andrade et al., 2020*). The infrastructure singularity of IoT, which unites the physical and virtual worlds and raises the personal risks inherent in both, is the main cause for concern. The question of privacy is another issue. Businesses want data collection to be transparent, including what is gathered and why, who can see it, and who controls access. There are also concerns about maintaining industry compliance standards and general safety issues involving equipment and users. The primary IoT security challenges are explained in the following section. Simply put, the more citations an article receives, the more influential it is in that field. Table 4 summarises smart city challenges and their advanced IoT-driven solutions.

## Security issues

IoT-enabled smart city infrastructures are becoming more and more vulnerable, as seen by recent cybersecurity incidents in 2023 and 2024. A significant DDoS attack that targeted a smart city in Asia in May 2024 exposed flaws in antiquated IoT protocols by interfering

**Table 4 Smart city challenges, and their advanced IoT-driven solutions.**

| Case study city | Key challenge | Advanced IoT-driven solution |
| --- | --- | --- |
| Masdar City | High energy consumption and reliance on non-renewable sources | AI-powered energy optimization, smart grid integration, and solar-powered microgrids |
| New York | Cybersecurity vulnerabilities in open data and critical infrastructure | AI-driven anomaly detection, blockchain-based cybersecurity frameworks, and zero-trust architecture |
| Copenhagen | Urban congestion, air pollution, and inefficient transport systems | IoT-enabled intelligent traffic systems, autonomous public transport, and smart cycling infrastructure |
| Singapore | Water scarcity and inefficient urban planning affecting sustainability | AI-enhanced water quality monitoring, real-time leak detection, and smart irrigation networks |
| Fujisawa | Waste management inefficiencies and limited renewable energy adoption | Decentralized waste-to-energy solutions, AI-powered recycling systems, and IoT-enabled waste monitoring |

with vital IoT systems in charge of public safety, utilities, and traffic control (*Alshahrani, 2023*). Similarly, hackers used digitalized power systems to launch a 70% increase in cyberattacks against U.S. utilities in 2024, raising fears about possible widespread disruptions to critical services (*Nirosha & Hemamalini, 2024*). Authorities in Amsterdam have delayed the rollout of smart traffic light projects because of privacy concerns, citing the dangers of GPS monitoring and mobile data collecting (*Richardson, 2024*). Ransomware attacks on critical smart city infrastructure, such as energy grids, transportation systems, and healthcare networks, have increased in recent years. Cybercriminals exploit IoT vulnerabilities to encrypt crucial city data and demand ransom payments, leading to service disruptions (*George, Baskar & Srikaanth, 2024*).

Many of Australia's 250,000 installed residential batteries were discovered to be susceptible to cyber assaults, exposing cybersecurity hazards in solar battery storage systems and igniting concerns about illegal system manipulation and dangers to the country's energy infrastructure (*Ahmadi-Assalemi et al., 2020*). These occurrences highlight the critical need for strong cybersecurity frameworks, AI-driven threat detection, and blockchain-based authentication solutions to protect IoT ecosystems from constantly changing cyber threats.

Security concerns continue to be crucial impediments to the widespread IoT acceptability and implementation. Consumers would hesitate to embrace IoT without assurance that their privacy is protected. IoT is extremely susceptible to cyberattacks due to these reasons (*Shah & Sengupta, 2020*):

- Frequently left unattended for long periods, and the objects are comparatively simple targets for physical harm.
- Since most communications are wireless, man-in-the-middle assaults represent a prevalent danger category to these systems.

The overall security of IoT may be jeopardized because exchanged messages are vulnerable to eavesdropping, malicious routing, message tampering, and other security issues.

**Peer**J Computer Science

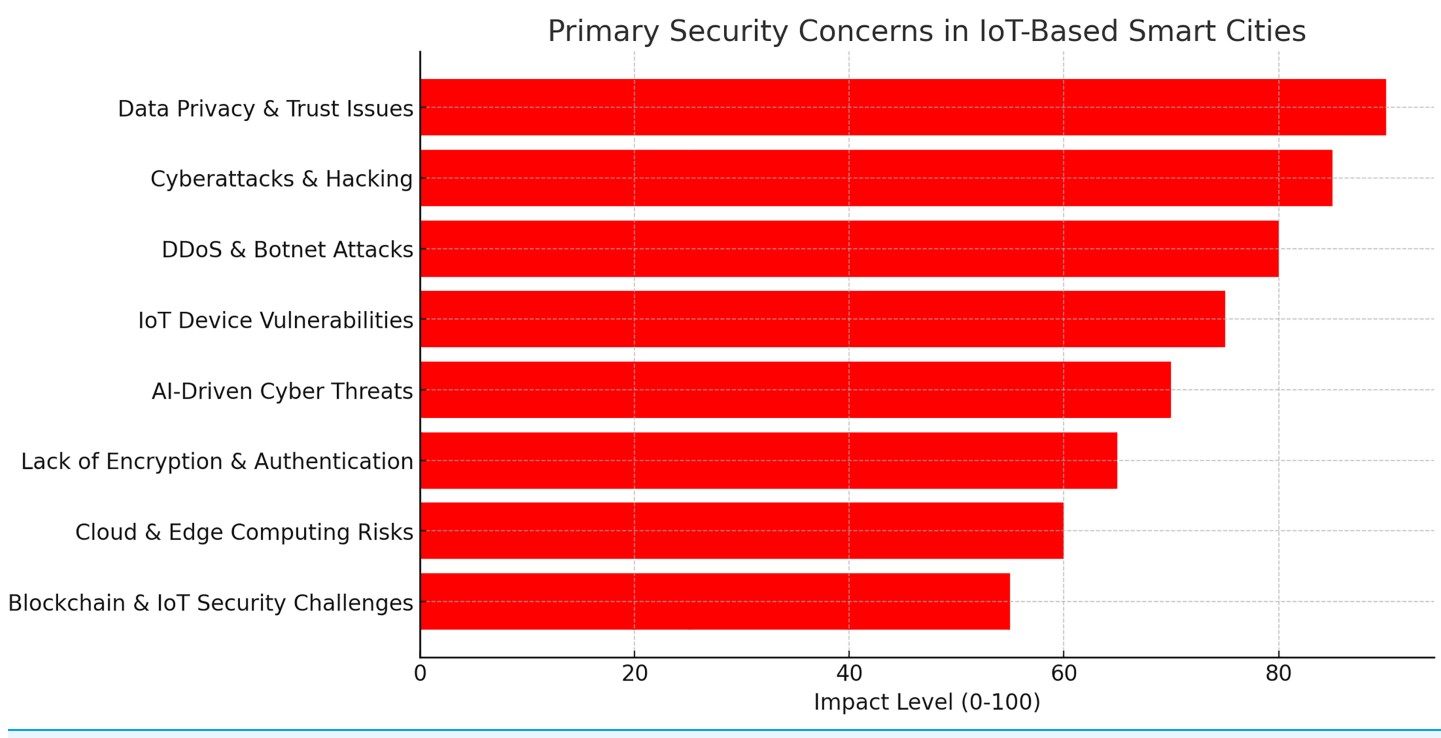

**Figure 4** The primary security concerns in IoT based smart cities.

- RFID tags have limited computational and energy resources, which makes it impossible for them to implement contemporary security measures.

Most of these vital IoT applications may suffer devastating effects if inaccurate or purposefully corrupted data are applied. The existing security mechanisms and algorithms may be relevant if the IoT devices have sufficient memory and computing power. Due to the resource limits of IoT items, conventional security solutions are too costly for IoT products. The following section elaborates on the main element that presents security challenges while implementing IoT-based smart cities. Figure 4 illustrates the primary security concerns in IoT based smart cities based on authors analysis.

**Privacy and trust**

The success of smart cities heavily depends on trust. To ensure that data is applied legally, ethically, and for more significant benefit, constructing a smart city seeks high standards of transparency and monitoring. Without sacrificing the efficacy or benefits to public safety, some measures (*e.g.*, General Data Protection Regulation (GDPR) can be implemented to ensure that privacy is respected. These IoT devices are weak points for attackers because frequently connected to public networks and can have unpatched software vulnerabilities. Besides, IoT data breaches can lead to financial loss, social unrest, and an increased political agenda. Given that IoT devices must be Internet-connected, one may claim that this is technology's most critical problem. By exploiting IoT devices as a means of access, cybercriminals are given a means of accessing confidential data and committing unlawful acts (*Haryono et al., 2024*).

**Data confidentiality, integrity and authentication issues**

Data confidentiality and authentication are other obstacles to smart city implementation, although most IoT devices disregard authentication. In configuring and managing a device, a subscriber can use a web browser to navigate to a specified IP address. Researchers have suggested using public-key encryption, where each object has a set of private and public keys. Each object retains its private key, and the base station holds all the public keys. Elliptic Curve Cryptography (ECC), NtruEncrypt, and Rabin's Scheme are the three leading suggestions for public key encryption algorithms suitable for IoT devices. The ECC offers goods capability without any complicated key management mechanism. The application of these algorithms to the IoT context is still being investigated. Moreover, it is unsuitable for all sorts of items, mainly RFID tags, where restricted resources remain a challenge. Trust issues plague public key encryption solutions. A base station with public keys, however, cannot establish that the objects are who claim to be (*Alhawamdeh et al., 2024*). To address concerns about data confidentiality, cities could adopt end-to-end encryption and implement transparent data usage policies that are accessible to the public.

**Data sharing and management**

In the cases of smart homes and smart healthcare, data sharing between the two sectors can result in several applications, such as intelligent logistics management, warehouse management, and individualized medical treatment. As most IoT implementations are based on diverse sensor types, the probability of exchanging erroneous, incorrect or inconsistent data is rather high. Such an issue has an impact on the models developed from the retrieved data. Hence, it is critical to assess the data quality upon collection. Data sharing issue encompasses multiple challenges that could influence the adoption of IoT. Although numerous cutting-edge security solutions exist in IoT, adopting blockchain has been recommended as an appropriate solution (*Byabazaire, O'Hare & Delaney, 2020*).

**Denial of service**

A distributed denial-of-service (DDoS) attack happens when all network devices unstably send unlimited messages, thus creating IoT network congestion and shutting down eventually. Cybercriminals execute DDoS attacks to control many compromised devices and halt important information from reaching its destination. The vast Internet device populations of cities provide cybercriminals with a practical attack vector. The security implications are enormous. For example, when thousands or ten thousand devices in a large metropolis communicate concurrently with users and among themselves, smart cities can become an ideal target for hackers looking to construct IoT bot networks. An IoT botnet refers to a collection of infected devices used to perform various activities without the knowledge of their authorized users (*Salim, Rathore & Park, 2020*). For instance, massive linked devices (mainly connected webcams made by the Chinese company XiongMai) saturated the infrastructure of Dyn Company in 2016, which led to a denial-of-service assault.

As a consequence of the attack, Dyn failed to offer DNS service. Some of the related objects used in the attack were brought about by Mirai malware. This program exploits drawbacks in some connected objects, such as the use of a default password that users have not changed. Moreover, cybercriminals increasingly employ IoT networks as a platform

for their attacks. Intrusion Detection Systems (IDSs) are the finest tools for detecting and preventing DoS and DDoS attacks (*Albasheer et al., 2022*). However, given the unique qualities and capacities of the objects, implementing such systems in an IoT infrastructure is indeed a tough challenge.

### Big data management

Communication technology is imminent in a smart city. A smart city becomes a source of enormous volumes of data, also known as big data, since the number of devices increases exponentially. Some programs can be used to control and assess city data while concurrently equipping users with helpful information. For example, IBM and the Brazilian government worked together to develop a city-wide instrumented system that gathers, processes, and analyses data flows from 30 sources, including weather data, traffic and public transportation data, data from emergency services, as well as data from municipal and utility services to name a few. Information is combined to analyze specific city life pacts and continually offer fresh answers (*Uddin & Hossan, 2024*).

### Network and transport issues

Network hacking occurs when an IoT device is infiltrated through its associated network. This security flaw allows the hacker to access and manipulate the IoT device. For instance, hackers manipulate the thermostat of an industrial furnace to ignite a fire or induce the collision of an autonomous vehicle by regulating its driving mechanisms. The transport layer is important to the IoT (*Tewari & Gupta, 2020*). Attacks on this layer and the routing system beneath it can significantly influence the network functions. The primary objectives of the transport layer are congestion control and end-to-end reliability. The Transmission Control Protocol (TCP) helps achieve these objectives in conventional networks. However, TCP is unsuitable for IoT due to the following reasons:

**Establishing the connection:** In TCP, the three-way handshake is the first step in the connection phase of each session. A minimal quantity of data will be transmitted among IoT devices. Hence, most of the session time is spent in the setup phase. This can result in increased energy and resource consumption.

**TCP makes end-to-end congestion control** possible. However, as most IoT communications are wireless, performance can be affected, making this setting unsuitable for TCP.

**Data buffering**: Both at the source and the destination, TCP stores data in a memory buffer. The first is at the source for the requirement of retransmission, and the second is at the destination for the necessity of ordered delivery. Such buffer management and allocation can be costly for objects. Thus, TCP cannot effectively be used for end-to-end transmission control in IoT. This demands the development of new transport layer protocol solutions.

### Cloud computing

Although many IoT-based smart city gadgets heavily rely on cloud services, the requirement for an internet connection to operate is indeed a significant challenge. Besides

not functioning while the network is operating slowly, it might also be synchronizing sensitive data or providing another possible entry point for IoT devices. Cloud-based technology can meet the bulk of the criteria for smart city applications. However, the physical distance between data collection and processing and the centrality of cloud computing shows that such solutions are inadequate to meet future requirements of smart city applications (especially real-time ones). Fog-based and edge computing options have been prescribed as answers to those issues (*Sadeeq et al., 2021*). By using a decentralized architecture and bringing the benefits and power of the cloud closer to where data are collected and used, fog computing increases the capabilities of cloud computing. The most logical and practical locations between the data source and the cloud are used for data distribution, processing, storage, and applications. Similar to this, edge computing moves data processing as close as possible to the data source to the edge of the network (*Sharma, Tomar & Hazra, 2024*).

## Heterogeneity issues

Recent studies on IoT can lead to a paradigm shift in comprehending the fundamental computer science ideas and standards for future way of life due to its heterogeneous nature (*Mishra & Agarwal, 2024*). The capacity for different linked object types to communicate with one another within a network is known as heterogeneity. The electronic devices typically have network-connected computers integrated and may differ regarding processing power, input-output capabilities, resource scale, connectivity technologies, and communication protocols. The three degrees of interoperability for heterogeneous systems are listed as follows:

- Interoperability with basic connectivity concentrates on the physical connections between devices.
- The administration of data communication is described in Network Interoperability.
- Syntactic interoperability's core focus is application-level interoperability. In IoT applications, data are gathered from several geographically separated items.

When data is retrieved from multiple protocols, the data formats become varied. These causes difficulty in efficiently assessing, process or storing data without a uniform format. Integrating data from several sources may be difficult due to poor standards. In order to facilitate effective and smooth data gathering among diverse IoT gadgets, it is crucial to define two important issues: (1) the standards for unified data encoding and (2) the information-sharing protocols.

## Infrastructures

Many companies are now offering IoT systems for smart cities, typically based on solutions created for pilot cities. While these platforms are frequently not ideal for other cities, most cities are concerned about vendor lock-in that could prevent them from using different vendors for future infrastructure and application improvements because everything must be tailored to the platform of that particular vendor. An infrastructure capable of handling such massive data globally effectively is not yet widely available (*Kumar et al., 2022*).

Technology that gathers and processes all sorts of data, such as sound or physical events, must be invented if smart cities are to function effectively. Smart cities and efficient use of massive volumes of data demand widespread wireless coverage and fast transmission rates, yet the requisite infrastructure is frequently lacking. Despite the expanding 5G coverage, it still does not guarantee reliable speeds. Millions of new cell towers should be erected to offer adequate network coverage. At present, each smart city has a unique appearance in terms of the infrastructure put in place, the available applications, and frequently even the IoT platforms applied. This signifies a barrier to developing a market for smart cities since it is difficult to reuse anything created for a town in another. On top of that, creating specialized platforms for each location is not financially feasible (*Kumar et al., 2022*).

## Government and finance

As massive investments are sought to build and sustain smart cities, all private and public resources at the municipal, state, and federal levels must work together. Due to the increasing number of parties with financial and intellectual stakes in smart project success, budget structuring, accountability, and access challenges have become even more complicated. Political cycles and their dynamics, as well as shifts in leadership or government priorities, can affect smart city development and operation and may eventually result in the postponement or restructuring of initiatives (*Hedegaard et al., 2024*). It is challenging to promote smart city projects when a unified viewpoint is required, mainly because government departments and agencies that can do so frequently run separately from one another. For example, selecting those in charge of regulating data protection or deciding how to distribute access to intelligent systems is challenging. I addition, integrating IoT platforms to enable information sharing, which can be realized by federating IoT systems and making some information accessible to this federation (*Byabazaire, O'Hare & Delaney, 2020*). Although such a setup is technically possible, the vast players involved and the federated system functioning across organizations and administration should share a common understanding to overcome arising challenges, such as defining roles and allocating budgets.

## Legislation and standards of different countries

Instead of the conventional functionality accessible *via* the user interface, many IoT devices ship with open ports to ease administration operations. For example, some passwords enable Telnet access using only an IP address. It is essential to reach an agreement on the interfaces and information modelling that Open & Agile Smart Cities (OASC) describes as the very minimum interoperability procedures (*Sung & Lu, 2024*). Besides, expenses can be decreased by using widely used open-source building blocks, such as those offered by FIWARE, as a base. Proprietary vendor components may be included in these. Imminently, vendor lock-in is avoided by using standards at the interface level. For instance, data from inside the home may be gathered *via* sensors on IoT-based home equipment. The interior sensors in a smart car may be used to track the driver's driving patterns. Using data mining techniques on this type of data enables identifying certain people. To effectively secure personal information and stop data-collection service

providers from abusing it, governments have begun passing strict regulations on protecting personal information. Moving away from monolithic proprietary platforms and toward modular platforms based on standards (*e.g.*, NGSI-LD and OMA Lightweight M2M) is the way to go (*Bauer, Sanchez & Song, 2021*).

## Sensors

Smart cities are made feasible by collecting data from sensors strategically placed throughout the city and on its inhabitants, analyzing them, and then mining them to improve the dwellers' quality of life. These sensors can assess the inner conditions of the smart city's infrastructure, such as its transit system, electrical grid, building condition and status, and movement of people. These data are transferred to the cloud to be mined and analyzed. However, some transmission and data usage issues have hindered the entire process's integrity, security, and confidentiality. However, such an issue is not unfounded because, in 2015, a cyberattack on the Ukrainian power infrastructure left 225,000 people without power (Case, 2016), bringing cyberattacks to the public's attention as a real threat. The 5G has been anticipated to be the key facilitator for IoT by enabling the connectivity of numerous sensors, supporting devices, and supporting actuators with severe computational and power restrictions. The modelling of such scenarios, particularly at the access network level and the specific technology combinations, cannot be escaped from future communication systems in their early phases of development.

## Radio frequency jamming

The idea of preventing a wireless device from connecting with other devices or a wireless network is called RF jamming. This is performed in settings requiring high levels of security, and all unwanted contact must be cut off. "RF Jammers" prevents communication between a device and other external devices. These RF jammers, which broadcast signals at the same frequency as the target device, are extremely powerful compared to the signals from the target device. By interfering with wireless communications to hinder their functionality, hackers can use wireless jamming to block wireless IoT devices. The RF jammer can cause IoT devices to lose connectivity and limit communication ability (*Anthi et al., 2024*). For instance, security alarms in smart cities can be jammed to enable an intruder to enter a premise due to a security breach. The combination of several signals with extremely high-power levels operating on the same frequency overwhelms the receiver, thus preventing it from decoding any target signal. Due to the widespread use of RF jamming in military applications, security toolmakers have also developed several strategies to combat it. However, most of those methods are confidential and not accessible to the public. There are some standard strategies for failing the RF jamming apparatus, such as using brief bursts of transmission with narrow bandwidth signals to make jamming difficult.

## LIMITATIONS AND FUTURE RESEARCH DIRECTIONS

This review's limitations should be considered when evaluating the outcomes. First, this article focuses on the literature review within the IoT sector by solely considering the

context of smart cities. Second, the minimum keyword occurrence criterion was deployed. Third, the root causes of the increase in publications regarding IoT in relation to smart cities are neglected in this study. A review of the literature and a close evaluation of the publications that laid the groundwork for this study stream unveiled several important gaps. Future research work should employ a mixed-method approach to analyze the role of multiple stakeholders in deploying IoT in smart cities. Besides, more studies are called to examine the adoption of smart cities at several levels, including the implementation level, the governance and management level, and the level of citizen cooperation. A deep investigation into the standardization of IoT devices is necessary, mainly because some IoT devices do not require authentication. This concerns security issues as a subscriber can navigate with a web browser to a specific IP address and control both the configuration and operation of the IoT device.

This study has examined a number of IoT-based smart city topics, emphasizing their main elements, uses, and difficulties. Even though IoT technologies offer revolutionary approaches to urban administration, interoperability issues, security flaws, and governance issues continue to be major obstacles to their broad implementation. Future studies should concentrate on a few crucial areas to improve the sustainability and resilience of smart cities enabled by the Internet of Things.

AI-driven IoT governance is one exciting avenue where AI may be used to predict urban demands, enhance system resilience against cyberattacks, and optimize decision-making processes. By improving real-time traffic monitoring, resource allocation, and predictive maintenance for smart infrastructure, AI-based algorithms can save operating costs and inefficiencies. Furthermore, AI-powered cybersecurity tools like automated threat mitigation and anomaly detection help fortify IoT networks against emerging cyber threats like ransomware and AI-driven assaults.

The creation of privacy-enhancing technologies (PETs) to guarantee safe data processing in smart city settings is another important research field. Novel encryption methods, homomorphic encryption, and federated learning models can assist in safeguarding user privacy while facilitating data-driven innovation, as IoT devices produce enormous volumes of sensitive and personal data. In order to ensure that smart city applications adhere to strict data protection rules like the General Data Protection Regulation (GDPR), research should also concentrate on creating regulatory frameworks that strike a balance between privacy protection and data accessibility (*Chiara, 2024*).

Furthermore, a worthwhile research topic is the function of blockchain in smart cities. By lowering dependency on centralized authority, blockchain-based solutions can enhance IoT security, data integrity, and decentralized identity management. By automating transactions in urban infrastructure, smart contracts can improve efficiency in domains like digital identity verification, energy trading, and transportation management. Future research should examine how blockchain may be integrated with AI and IoT to create autonomous, tamper-proof smart city ecosystems that improve security and transparency.

Future studies should investigate blockchain-based solutions for decentralized smart city frameworks, privacy-enhancing technologies for safe data management, and AI-driven governance for intelligent decision-making. In order to ensure that smart cities

live up to their promise of improving urban living while protecting privacy and security, academics can help create more secure, effective, and resilient IoT-powered urban environments by tackling these issues.

## CONCLUSIONS

In conclusion, this survey article has explored various aspects of IoT-based smart cities, focusing on their primary components, applications, and challenges. We investigated the diverse landscape of smart homes, health, industry, energy, infrastructure, city services, agriculture, and waste management within the framework of IoT. Additionally, we delved into the crucial role of AI in enhancing IoT capabilities and examined various sensing technologies and wireless communication methods vital for smart city implementations.

Furthermore, this study presented real-world synergies and case studies from cities like Masdar City in Abu Dhabi, New York, Copenhagen, Singapore, Fujisawa, and Kashiwa-no-ha in Japan. While highlighting the promising advancements, we also addressed the primary challenges associated with adopting IoT in smart cities, including security issues, big data management, networking concerns, and governance issues. Recognizing the importance of overcoming these challenges, we outlined potential limitations and proposed future research directions. This survey acknowledges the critical need to address security and privacy concerns, ensure data integrity, and establish standards for global IoT implementations.

### Funding

This work was supported by the Deanship of Research and Graduate Studies at King Khalid University through the Large Research Project under grant number (RGP 2/291/45). The funders had no role in study design, data collection and analysis, decision to publish, or preparation of the manuscript.

### Grant Disclosures

The following grant information was disclosed by the authors:
Deanship of Research and Graduate Studies at King Khalid University: RGP 2/291/45.

### Competing Interests

The authors declare that they have no competing interests.

### Author Contributions

- Sayeed Salih conceived and designed the experiments, analyzed the data, performed the computation work, prepared figures and/or tables, authored or reviewed drafts of the article, and approved the final draft.
- Abdelzahir Abdelmaboud conceived and designed the experiments, performed the experiments, analyzed the data, performed the computation work, prepared figures and/or tables, authored or reviewed drafts of the article, and approved the final draft.

- Omayma Husain performed the experiments, performed the computation work, prepared figures and/or tables, and approved the final draft.
- Abdelwahed Motwakel conceived and designed the experiments, authored or reviewed drafts of the article, and approved the final draft.
- Hashim Elshafie conceived and designed the experiments, analyzed the data, authored or reviewed drafts of the article, and approved the final draft.
- Mahir Sharif conceived and designed the experiments, authored or reviewed drafts of the article, and approved the final draft.
- Mosab Hamdan performed the experiments, analyzed the data, performed the computation work, prepared figures and/or tables, authored or reviewed drafts of the article, and approved the final draft.

## Data Availability

The statistical data is available in the Supplemental File.

## Supplemental Information

Supplemental information for this article can be found online at http://dx.doi.org/10.7717/peerj-cs.2816#supplemental-information.

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
