# Peer review of "IoT in urban development: insight into smart city applications, case studies, challenges, and future prospects"

_PeerJ Computer Science, doi:10.7717/peerj-cs.2816_

## Round 0.1 · original submission · Major Revisions

· Academic Editor

Major Revisions

The paper provides valuable insights into IoT applications in smart cities, but major revisions are needed to improve clarity, methodological rigour, and empirical support. The introduction should better define the research gap and the study’s contributions. The literature review needs updates with more recent sources. The methodology lacks detail on case study selection and how techniques have been applied. Figures, tables, and charts should be better integrated and explained. The discussion on security challenges should be expanded. Finally, careful proofreading is required to fix grammar, formatting, and inconsistencies. The revisions suggested by the reviewers will strengthen the paper.

·

Basic reporting

To some extent

Experimental design

Literature only presented

Validity of the findings

No comparisions

Additional comments

A. The contributions listed in the introduction section in bullets should be combined into only four bullets
B. The heading "Paper Organization" should be removed and the paragraph of this heading should remain and should be reduced
C. The analysis, such as why this study has importance, what the existing literature has limitations, and what recommendations are required, is missing.
D. Only a few tables are not sufficient to present the work done. There should be proper representations
E. Charts should also be part of the paper

·

Basic reporting

Line 121: The current sentence reads:

"Smart cities require rigorous adherence to a precise set of standards that enables numerous systems
to operate, integrate, and harmonize with one another."

Suggestion: Break this sentence into two for better readability. For example:

"Smart cities need to adhere to rigorous standards. These standards enable various systems to
operate, integrate, and harmonize effectively."

This restructuring makes the point clearer and easier to follow.

In Lines 57-86: In this section, you discuss the general concept of IoT in smart cities but don't fully address the knowledge gap your study is attempting to fill. The paragraph currently outlines the growth of smart cities and IoT technologies without clearly stating:

What specific gaps in the current literature exist?

How your study address these gaps (e.g., through a unique methodology, case study selection, or focus on under-researched challenges like privacy or governance).

Suggested revision:

Add a statement such as:

"While previous studies have broadly explored IoT applications in smart cities, they often lack a detailed examination of security vulnerabilities and their mitigation. This study addresses these gaps by analyzing case studies of IoT implementation with a focus on identifying practical solutions to security and scalability challenges."

Expanding on this aspect will make your research’s purpose and contribution more compelling.



Please ensure some of the titles in your references are Titlevcae and Not ALL Uppercase. Check lime 1245 & 1246

Experimental design

Lines 207–214 (Challenges Section):

The section mentions strategies like encryption and blockchain but lacks depth in explaining how these strategies are applied.


I will suggest you add some additions

"For example, blockchain can secure IoT data by creating immutable transaction logs that prevent unauthorized alterations. This technology can be particularly effective in applications like smart healthcare and waste management, where data integrity is critical."

Validity of the findings

Lines 780–812 (Challenges for IoT Adoption):

The discussion of challenges would benefit from including specific recommendations for unresolved issues.

You could say:

"To address concerns about data confidentiality, cities could adopt end-to-end encryption and implement transparent data usage policies that are accessible to the public."

Additional comments

The manuscript covers an impressive range of IoT applications, supported by real-world case studies. The emphasis on both technical and social aspects of IoT adoption is commendable.

Reviewer 3 ·

Basic reporting

- Language and Clarity
The paper is well-written, with clear, unambiguous, and professional language throughout. However, some sections, particularly in the introduction and methodology, contain overly long and complex sentences that could be revised for better readability.
There are minor grammatical errors and awkward phrasings that could be improved. Example:
"Through the integration of sensors, cell phones, AI, data analytics, and cloud computing, smart cities worldwide are evolving to be more efficient, productive, and responsive to their residents' needs."
This could be rewritten as: "By integrating sensors, mobile devices, AI, data analytics, and cloud computing, smart cities are becoming more efficient and responsive to residents' needs."

- Literature References & Context
The paper provides a well-rounded literature review, citing relevant sources on smart cities, IoT components, and challenges.
However, some references are outdated. For example, the reliance on sources from 2014-2016 (e.g., Zanella et al., 2014; Pan & McElhannon, 2017) might not reflect the latest technological advancements in IoT security, AI-driven urban solutions, and regulatory frameworks.
The authors should incorporate more recent works (post-2020) to reflect the latest developments in the field.

- Structure & Figures/Tables
The paper follows a professional structure with clear sections, headings, and subheadings.
Figures and tables are present and well-labeled, but they lack a proper explanation within the text.
Suggestion: Each figure should be explicitly referenced and briefly described in the surrounding text to enhance clarity.
The raw data supporting the case studies is not clearly presented. If available, sharing datasets or detailed statistics would enhance credibility.

- Scope and Contribution
The paper aligns with the journal’s scope, as it presents a cross-disciplinary analysis of IoT applications in urban development.
The introduction provides sufficient background and motivation, but it could benefit from a more explicit problem statement.
Suggestion: The introduction should more clearly define the research gaps and objectives.

Experimental design

- Survey Methodology & Investigation Rigor
The research adopts a qualitative methodology, relying on literature synthesis and case studies. While this is suitable for a review paper, the methodology section lacks explicit criteria for selecting case studies.
Suggestion: Clearly define how case studies were chosen (e.g., by geographic diversity, technological maturity, or policy frameworks).
The study does not appear to include primary data collection or systematic meta-analysis, which would have strengthened the findings.
Suggestion: If feasible, integrating statistical comparisons of IoT adoption across different cities would enhance the study’s depth.

- Source Citation and Logical Organization
The paper properly cites sources, but direct quotations are minimal. Paraphrased content is well-integrated.
The logical flow of sections is generally clear, but some sections could be better connected. For example:
The "Challenges and Security Considerations" section should directly follow the discussion of IoT applications to maintain coherence.

Validity of the findings

- Argument and Evidence Support
The paper provides a well-structured argument regarding IoT's role in urban development.
The authors make strong claims about IoT improving urban infrastructure but do not always provide empirical evidence.
Example: The statement "IoT will revolutionize urban mobility and reduce traffic congestion by 50%" lacks a supporting citation.
Suggestion: Provide concrete case study results or refer to quantitative data from existing smart city projects.

- Areas for Improvement:
Needs a stronger methodological framework for case study selection.
Should incorporate more recent literature and empirical data.
Some figures/tables need better integration and explanation.
The discussion on security challenges could be expanded to cover recent cybersecurity trends (e.g., ransomware, AI-driven cyber threats).

- Conclusions and Future Directions
The conclusion summarizes key findings effectively, but it does not sufficiently address gaps or unresolved challenges.
Suggestion: Expand on potential research directions, such as AI-driven IoT governance, privacy-enhancing technologies, or the role of blockchain in smart cities.
The paper could also include a comparative table summarizing the challenges and corresponding solutions across different case studies.

Additional comments

- Strengths of the Manuscript:
Provides a comprehensive literature review on IoT in urban development.
Offers a detailed discussion of multiple smart city case studies.
Clearly outlines key challenges and future prospects.

---

## Round 0.2 · Minor Revisions

· Academic Editor

Minor Revisions

Overall, all comments have been carefully considered. Some of the remarks from Reviewer 2 require a minor revision.

Consider also the following minor adjustments. Replace "our" (e.g., "study") with "this" and avoid using first-person pronouns.

Revise sentences at lines 822-825: the first one is just a fragment; in the second one, there is an incorrect use of the semicolon. Also, the citation is repetitive.

Sentences in lines 830-832 present some issues. Please review the entire Case Studies Discussion that has been added during the review and, more generally, the whole document to ensure all sentences are correct. For instance, at line 167, a verb is missing, and at line 56, a parenthesis is missing.

·

Basic reporting

The phrase "By integrating sensors, mobile devices, AI, data analytics, and cloud computing, smart cities are becoming more efficient and responsive to residents' needs." (Lines 29-31) was revised.

The authors have done a commendable job in addressing the reviewers’ comments. Most of the necessary corrections have been made, significantly improving the manuscript's clarity, structure, and integration of figures and tables. The inclusion of more recent references strengthens the background and relevance of the study. Overall, the revisions are well-executed, and the manuscript is much improved.

Experimental design

The case study selection process has been described in detail in the “Comparative Case Study Analysis” (lines 233-248), but ensure consistency in the selection criteria (geographic diversity, technological maturity, policy frameworks).

If possible, provide a clearer justification for choosing Masdar City, New York, Copenhagen, Singapore, and Fujisawa over other cities.

Great work don here

Validity of the findings

No comment

Reviewer 3 ·

Basic reporting

no comment

Experimental design

no comment

Validity of the findings

no comment

---

## Round 0.3 · accepted · Accept

· Academic Editor

Accept

The authors have addressed all of the reviewers' comments. I have assessed the revision myself and consider the manuscript ready for publication.